# I2VCONTROL-CAMERA: PRECISE VIDEO CAMERA CONTROL WITH ADJUSTABLE MOTION STRENGTH

**Wanquan Feng**[1✉]    **Jiawei Liu**[1]    **Pengqi Tu**[1]    **Tianhao Qi**[1,2]    **Mingzhen Sun**[1,3]
**Tianxiang Ma**[1]    **Songtao Zhao**[1]    **Siyu Zhou**[1]    **Qian He**[1]

[1]ByteDance China    [2]University of Science and Technology of China (USTC)
[3]Institute of Automation, Chinese Academy of Sciences (CASIA)

## ABSTRACT

Video generation technologies are developing rapidly and have broad potential applications. Among these technologies, camera control is crucial for generating professional-quality videos that accurately meet user expectations. However, existing camera control methods still suffer from several limitations, including control precision and the neglect of the control for subject motion dynamics. In this work, we propose **I2VControl-Camera**, a novel camera control method that significantly enhances controllability while providing adjustability over the strength of subject motion. To improve control precision, we employ point trajectory in the camera coordinate system instead of only extrinsic matrix information as our control signal. To accurately control and adjust the strength of subject motion, we explicitly model the higher-order components of the video trajectory expansion, not merely the linear terms, and design an operator that effectively represents the motion strength. We use an adapter architecture that is independent of the base model structure. Experiments on static and dynamic scenes show that our framework outperformances previous methods both quantitatively and qualitatively. Project page: https://wanquanf.github.io/I2VControlCamera.

## 1 INTRODUCTION

Video generation technologies are explored to synthesize dynamic and coherent visual content, conditioned on various modalities including text (Blattmann et al., 2023c; Wang et al., 2024a; Gupta et al., 2023) and images (Blattmann et al., 2023b; Feng et al., 2024). Video generation has broad application potential across various fields, such as entertainment, social media, and film production. Motion controllability is crucial for ensuring that generated videos accurately meet user expectations, with camera control being one of the most important aspects. Camera control is the process of adjusting the position, angle, and motion of a camera, resulting in changes to the composition, perspective, and dynamic effects of a video. This technique is essential for generating professional-quality videos, as it influences the attention of viewers and enhances the expressiveness of scenes.

Although precise camera control is crucial for producing high-quality videos, existing methods still face challenges. The first challenge pertains to the precision and stability of control. The lack of precision would result in an inaccurate reflection of the user control intention, significantly degrading user satisfaction. The second challenge is ensuring the natural dynamics of the subjects themselves, independent of camera movements. Similar to the challenges in multi-view (Mildenhall et al., 2020; Kerbl et al., 2023) and 3D geometric algorithms (Wang et al., 2021), where static scenes are much easier to handle than dynamic ones (Pumarola et al., 2020; Cai et al., 2022), generating plausible dynamics in videos proves to be more complex than managing static elements.

While AnimateDiff (Guo et al., 2024b) utilizes LoRA (Hu et al., 2022) strategy for controlling camera movements, the motion-LoRAs are confined to a limited set of fixed movement modes, lacking flexibility, and it only allows for coarse control, thus failing to provide precise scale adjustments. A direct and intuitive approach allowing for arbitrary camera movements is embedding the camera pose matrix, as in MotionCtrl (Wang et al., 2023). However, this method results in sparse input signals that heavily rely on the training set distribution, which leads to poor generalization capability. Consequently, it may inadequately respond to less common camera parameters within the training

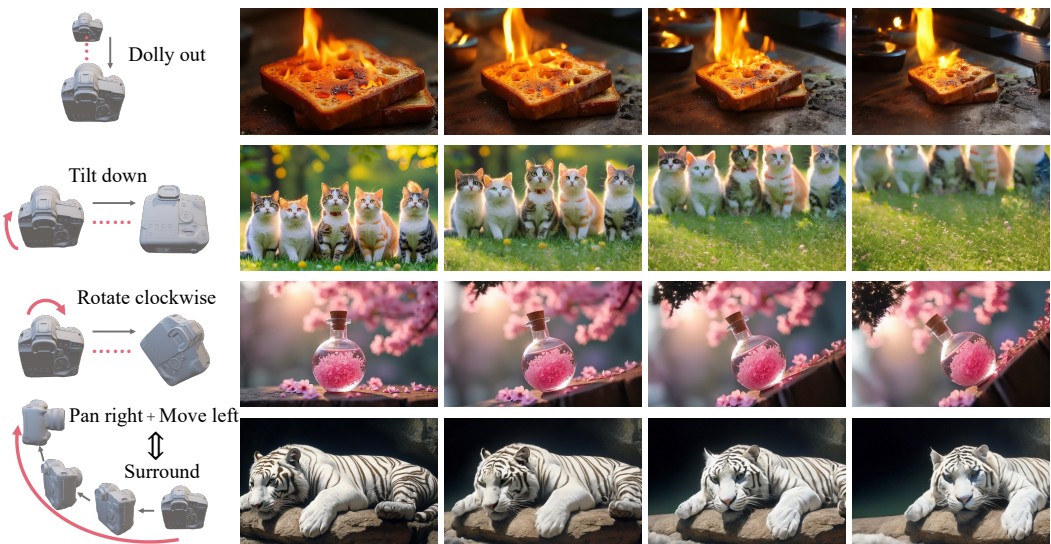

Figure 1: We propose **I2VControl-Camera**, a novel camera control method for image-to-video generation, offering high control precision and adjustable motion strength.

dataset, and thus hinders precise control over the motion's amplitude. Although CameraCtrl (He et al., 2024) attempts to mitigate this sparsity issue by employing Plücker embeddings (Sitzmann et al., 2021), this parameterization lacks information of the input image, and it actually does not offer any additional information compared to the camera matrix used in MotionCtrl. Another natural strategy is novel view synthesis, which uses 3D implicit representations that can be rendered from arbitrary views, such as Cat3D (Gao* et al., 2024). Unfortunately, this strategy cannot support subject motion well, thus undermining the core goal of creating dynamic video content.

In this paper, we propose **I2VControl-Camera**, a camera control method (some examples shown in Fig. 1) to surmount these prevalent issues in image-to-video generation, enhancing the control precision and adding control over the dynamic strength of subject motion in video output. To ensure control precision and stability, we use point trajectories in the camera coordinate system as our control signals, instead of extrinsic matrix. From the point trajectory function, we extract the linear term to serve as a proxy for camera control, ensuring high precision, stability and user friendliness. To control the motion strength, we further represent object motions with higher-order terms in the trajectory function and explicitly model the degree of dynamics. Specifically, we employ the derivative of the high-order terms to compute the motion speed of each point and integrate them in the image domain to obtain the entire motion strength as the control input of the network. This approach allows us to accurately gauge and adjust the amplitude of subject motion dynamics.

We construct training data from regular RGB videos registering 3D tracking information and motion mask for them. Our approach features an adapter architecture that remains agnostic to the underlying base model structure. Experimentally, we conduct experiments in both static and dynamic scenes. For static scenes, we can set the motion strength to zero, resulting in significantly higher precision than previous methods. In dynamic scenes, we can configure a higher motion strength, which allows for both high control precision and vivid subject motion. Our approach outperforms previous methods both quantitatively and qualitatively. In summary, our **contributions** include:

- We explicitly model decoupled motion representations: 3D rigid point trajectories and motion strength for camera and subject motion controls.
- We propose a data pipeline to construct training control signals from RGB videos.
- For both static and dynamic scenes, our method outperformances previous methods both quantitatively and qualitatively.

## 2 RELATED WORK

### 2.1 TEXT TO VIDEO SYNTHESIS

Text-to-video generation requires models to synthesize realistic videos based on given textual descriptions. Recent progress in diffusion models has boosted the quality of T2V generation to an

unprecedented degree, achieving both impressive visual quality and surprising text-video consistency (Brooks et al., 2024; Blattmann et al., 2023b). Image Video (Ho et al., 2022) cascaded multiple video generation and super-resolution diffusion models to generate long and high-resolution videos from textual descriptions. Make-A-Video (Singer et al., 2022) extended a diffusion-based T2I model to T2V in a spatiotemporal factorized manner. Based on the successful experiences of image generation methods, several works (Wang et al., 2024a; Girdhar et al., 2023; Mei & Patel, 2023) performed T2V by first generating images from texts and then synthesizing videos based on images. EMU VIDEO (Girdhar et al., 2023) introduced adjusted noise schedules and a multi-stage training strategy for high-quality video generation. To reduce the computational complexity of video generation, other works (Blattmann et al., 2023c; He et al., 2022; Yu et al., 2023; Gupta et al., 2023) explored different designs of video auto-encoders, which can map a high-dimensional video into a low-dimensional latent space. LVDM (He et al., 2022) compressed videos from both the spatial and temporal dimensions, obtaining a low-dimensional 3D latent for each video. In addition, Lumiere (Bar-Tal et al., 2024) and Latte (Ma et al., 2024) explored different 3D model structures. Recently, Sora (Brooks et al., 2024) showed the power of DiT (Peebles & Xie, 2022) in T2V task.

## 2.2 IMAGE TO VIDEO SYNTHESIS

Image-to-video task aims to generate videos with a static image as the condition. One classic strategy is integrating CLIP embeddings of the static image into DPMs. For instance, VideoCrafter1 (Chen et al., 2023a) and I2V-Adapter (Guo et al., 2024a) utilized a dual cross-attention layer, similar to the IP-Adapter (Ye et al., 2023), to fuse these embeddings effectively. However, due to the notorious issue of CLIP image encoders losing fine-grained details, subsequent works (Hu, 2024; Wei et al., 2024) have proposed using more expressive image encoders to capture finer image features. In addition, another strategy is to expand the input channels of DPMs to concatenate noisy frames and the static image. Notable works in this category include SEINE (Chen et al., 2023b), Pixel-Dance (Zeng et al., 2024), AnimateAnything (Dai et al., 2023), and PIA (Zhang et al., 2024), which have demonstrated superior results by enhancing the input channels to integrate image information more effectively. Finally, methods such as DynamiCrafter (Xing et al., 2023), I2VGen-XL (Zhang et al., 2023), and SVD (Blattmann et al., 2023a) combined channel concatenation and attention mechanisms to simultaneously inject image features, aiming to achieve consistency in both global semantics and fine-grained details. This dual approach ensured that the generated videos maintained a high level of fidelity to the original static images while introducing realistic and coherent motion.

## 2.3 VIDEO CAMERA CONTROL

While methods aiming to control video foundation models continue to emerge, relatively few works explore how to manipulate camera motions in generated videos. AnimateDiff (Guo et al., 2024b) employed temporal motion LoRA (Hu et al., 2022) trained on video datasets with similar camera motions, where one single trained LoRA can control a specific type of camera motion. MotionCtrl (Wang et al., 2023) proposed to employ an adaptor structure to encode the extrinsic matrix of each frame into the temporal attention layers. Further, CamereCtrl (He et al., 2024) utilized the Plücker embedding to improve the controllability. Camtrol (Hou et al., 2024) proposed a simple training-free method to directly render static point cloud to multiview frames and construct the final output video in a video-to-video manner. CamCo (Xu et al., 2024) integrated an epipolar attention module in each attention block that enforces epipolar constraints to the feature maps, which keeps 3D-consistent well but causes small motion dynamic. In our work, we propose a method that can enhance the precision of camera control and add the control over subject motion strength.

## 3 METHOD

### 3.1 VIDEO REPRESENTATION AND NOTATIONS

In this section, we introduce the video representation and notation used in this paper. First, we stipulate that the coordinates of all points we study are in the camera coordinate system. Although both the camera and the captured scene may move, we transfer all dynamics to the camera coordinate system, as in Fig. 2. Intuitively, the entire 3D world can be divided into the the static part and the dynamic part, where the static part corresponds to a linear motion in the camera coordinate system.

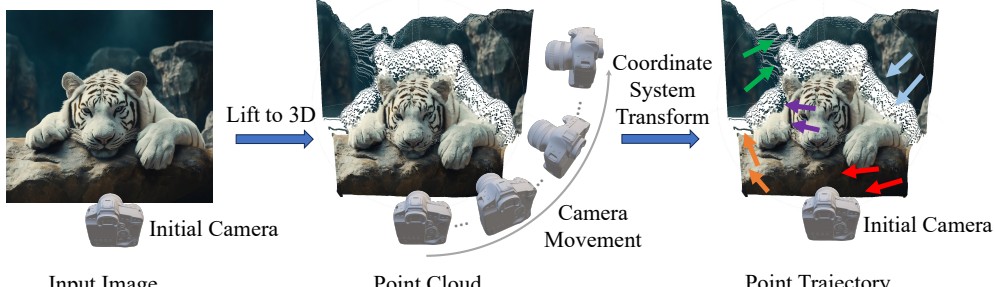

Figure 2: We lift the input image from 2D to 3D as a RGBD point cloud. When the camera moves, the 3D points can be considered as moving in the camera coordinate system. Then we project them onto 2D according to current camera pose to obtain the 2D point trajectory.

Consider a dynamic sequence $\mathcal{F}(\mathbf{p}, \lambda)$:

$$\mathcal{F}(\mathbf{p}, \lambda) : \mathbb{R}^3 \times [0, \Lambda] \to \mathbb{R}^3, \text{ s.t. } \mathcal{F}(\mathbf{p}, 0) = \mathbf{p} \tag{1}$$

where $\lambda \in [0, \Lambda]$ represents a time moment during the video, and $\mathbf{p} \in \mathbb{R}^3$ denotes a point of the entire 3D world. Notice that we specifically enforce $\mathcal{F}(\mathbf{p}, 0) = \mathbf{p}$, to ensure that $\mathcal{F}$ accurately defines the 3D motion trajectory originating from the first frame. Considering the physical properties of the macroscopic world, it is reasonable to consider $\mathcal{F}$ as a smooth mapping function. Naturally, we can assert that for any given $\lambda \in [0, \Lambda]$, there exist unique $\mathbf{R}_\lambda \in \mathbb{R}^{3 \times 3}$ and $\mathbf{t}_\lambda \in \mathbb{R}^3$ such that:

$$\mathcal{F}(\mathbf{p}, \lambda) = \mathbf{R}_\lambda \cdot \mathcal{F}(\mathbf{p}, 0) + \mathbf{t}_\lambda + o(\mathbf{p}), \tag{2}$$

where $o(\mathbf{p})$ denotes an infinitesimal of higher order than $\mathbf{p}$. To simply prove it, we only need to perform a Maclaurin expansion of $\mathcal{F}(\mathbf{p}, \lambda)$ and $\mathcal{F}(\mathbf{p}, 0)$ at $\mathbf{p} = \mathbf{0}$:

$$\mathcal{F}(\mathbf{p}, \lambda) = \mathcal{F}(\mathbf{0}, \lambda) + \mathbf{J}_\mathcal{F}(\mathbf{0}, \lambda) \cdot \mathbf{p} + o(\mathbf{p}), \tag{3}$$
$$\mathcal{F}(\mathbf{p}, 0) = \mathcal{F}(\mathbf{0}, 0) + \mathbf{J}_\mathcal{F}(\mathbf{0}, 0) \cdot \mathbf{p} + o(\mathbf{p}), \tag{4}$$

where $\mathbf{J}_\mathcal{F}$ denotes the Jacobian matrix, representing the gradient for vector-valued functions. Subtracting the two equations and performing a simple calculation yields:

$$\mathcal{F}(\mathbf{p}, \lambda) = (\mathbf{I} + \mathbf{J}_\mathcal{F}(\mathbf{0}, \lambda) - \mathbf{J}_\mathcal{F}(\mathbf{0}, 0)) \cdot \mathcal{F}(\mathbf{p}, 0) + \mathcal{F}(\mathbf{0}, \lambda) + o(\mathbf{p}). \tag{5}$$

Evidently, we can define:

$$\mathbf{R}_\lambda \triangleq \mathbf{I} + \mathbf{J}_\mathcal{F}(\mathbf{0}, \lambda) - \mathbf{J}_\mathcal{F}(\mathbf{0}, 0), \mathbf{t}_\lambda \triangleq \mathcal{F}(\mathbf{0}, \lambda), \tag{6}$$

Subsequently, we further denote:

$$\mathcal{G}(\mathbf{p}, \lambda) \triangleq \mathcal{F}(\mathbf{p}, \lambda) - (\mathbf{R}_\lambda \cdot \mathcal{F}(\mathbf{p}, 0) + \mathbf{t}_\lambda) = o(\mathbf{p}), \tag{7}$$

which actually represents the extent of nonlinearity, being a higher-order infinitesimal with respect to $\mathbf{p}$ than the linear term. Up to now, we have introduced the variables $(\mathbf{R}_\lambda, \mathbf{t}_\lambda, \mathcal{G}(\mathbf{p}, \lambda))$ to facilitate our forthcoming discussion on video camera control.

## 3.2 CONTROL SIGNAL CONSTRUCTION

While the most intuitive method is to directly employ $\mathbf{R}_\lambda$ and $\mathbf{t}_\lambda$ as the control signals, we aim to overcome the previously mentioned challenges of controllability and subject motion. Denote the region of 3D points captured by the first frame as $\Omega \subseteq \mathbb{R}^3$. We compute the linear translation for $\Omega$ and project it to 2D, which defines a **point trajectory** on the camera plane:

$$\mathbf{T}_\lambda = \Pi(\mathbf{R}_\lambda \cdot \Omega + \mathbf{t}_\lambda), \lambda \in [0, \Lambda] \tag{8}$$

where $\Pi$ is the projection operation. Compared to $\mathbf{R}_\lambda$ and $\mathbf{t}_\lambda$, $\mathbf{T}_\lambda$ offers a denser representation, thereby providing enhanced controllability and stability.

However, at the same time, this could further inhibit the motion of the nonlinear parts, which is undesirable. To address this issue, we proceed to model the motion of the nonlinear parts (dynamic regions in the world system) as well. Considering that we have already defined the variable $\mathcal{G}(\mathbf{p}, \lambda)$ to measure the extent of nonlinearity, we employ its first-order derivative with respect to time $\lambda$ to quantify the degree of motion dynamics at time moment $\lambda$. In our generative tasks, we cannot control

the motion of every individual point, so we instead resort to a secondary strategy. We integrate the $L_2$ norm of the first-order derivative (physically represents the motion speed of the point, as shown in Fig.3) across the entire domain $\Omega$, to characterize an overall measure of **motion strength**:

$$m_\lambda = \frac{1}{|\Omega|} \int_\Omega \|\frac{\partial \mathcal{G}(\mathbf{p}, \lambda)}{\partial \lambda}\|_2 \, d\mathbf{p} = \frac{1}{|\Omega|} \int_\Omega \sqrt{\left(\frac{\partial \mathcal{G}(\mathbf{p}, \lambda)}{\partial \lambda}\right)^T \cdot \left(\frac{\partial \mathcal{G}(\mathbf{p}, \lambda)}{\partial \lambda}\right)} \, d\mathbf{p} \tag{9}$$

Up to this point, we have fully defined the inputs of our camera control framework, $(\mathbf{T}_\lambda, m_\lambda)$, where the former enhances controllability and stability, and the latter effectively represents the extent of motion dynamics, thus fulfilling the original intent of our designed method.

In addition, we discuss some properties for the control signals. As discussed in Sec. 3.1, $\Omega$ can be divided into a static part and a dynamic part, which we can denote as $\Omega_S$ and $\Omega_D$ respectively. Notably, $\Omega_S$ corresponds to the linear motion within the camera system. In fact, we have:

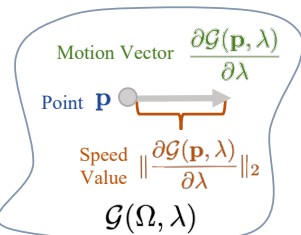

$$\mathcal{F}(\mathbf{p}, \lambda) \equiv \mathbf{R}_\lambda \cdot \mathcal{F}(\mathbf{p}, 0) + \mathbf{t}_\lambda, \forall \mathbf{p} \in \Omega_S. \tag{10}$$

In other words, $\mathcal{G}(\mathbf{p}, \lambda) \equiv 0$ on $\Omega_S$. According to Eq. 10, if we can obtain the partition $\Omega = \Omega_S \sqcup \Omega_D$, we can calculate $(\mathbf{T}_\lambda, m_\lambda)$ by simply linear fitting the point trajectory function $\mathcal{F}(\mathbf{p}, \lambda)$.

Figure 3: Illustration of motion strength (speed value).

### 3.3 DATA PIPELINE

In Sec. 3.2, we theoretically analyzed how to derive the input signal $(\mathbf{T}_\lambda, m_\lambda)$ for camera control. In this section, we show how to compute them for the real-world video data $\mathbf{V}_{gt}$. For the real-world video, the timesteps is a discrete sequence $\lambda \in [0, T] \cap \mathbb{Z}$, where $\lambda$ represents the timestep index. The region captured by the first frame can be organized on $H \times W$ pixels, denoted as $\Omega = \{\mathbf{p}_{ij}\}_{i,j=1}^{H,W}$. Further, we divide the whole point set $\{\mathbf{p}_{ij}\}_{i,j=1}^{H,W}$ into the static part and the dynamic part:

$$\Omega = \{\mathbf{p}_{ij}\}_{i,j=1}^{H,W} = \Omega_S \sqcup \Omega_D, \tag{11}$$

where $\Omega_S$ denotes the static part, and $\Omega_D$ denotes the dynamic part. Different from the theoretical analysis discussed in above sections, there are several major gaps between real-world RGB video data and the continous trajectory function:

- **Lack of 3D Information:** Real-world video data only contains 2D pixels without 3D information.

- **Lack of Temporal Correspondence:** The raw video data does not explicitly involve the information about the temporal movement of dense points as described by the continous trajectory function.

- **Lack of dynamic/static partition:** In real-world video data, discerning which regions are dynamic and which are static remains ambiguous, especially when the camera itself is also mobile. This introduces a coupling between the movement of objects and the motion of the camera.

To address the first issue, we employ metric depth estimation method, Unidepth (Piccinelli et al., 2024), to bridge the gap between 2D and 3D data representation. For the second issue, we utilize a tracking method, SpatialTracker (Xiao et al., 2024), to establish pixel correspondence between consecutive frames, so that we can obtain the discrete trajectory (still denoted as $\mathcal{F}(\mathbf{p}, \lambda)$ for ease of reading). For the third issue, we need to extract $\Omega_S, \Omega_D \subseteq \Omega$ from $\Omega$. A key insight lies in Eq. 10, which means the trajectory on $\Omega_S$ can be linearly fitted well. We solve this problem in a iterative manner, as described in Alg. 1.

Iteratively, we fit the trajectory and extract the well-fitted region as the updated static region $\Omega_S$, while the remaining part is the dynamic part $\Omega_D$. Once we obtain the result $\Omega_S$ and $\Omega_D$, like in each iteration, we can final compute $(\mathbf{R}_\lambda, \mathbf{t}_\lambda)$ by addressing a nonlinear least squares problem using the L-BFGS (Liu & Nocedal, 1989) algorithm:

$$(\mathbf{R}_\lambda, \mathbf{t}_\lambda) = \arg\min_{\mathbf{R}, \mathbf{t}} \|\Pi(\mathcal{F}(\Omega_S, \lambda)) - \Pi(\mathbf{R} \cdot \Omega_S + \mathbf{t})\|^2. \tag{12}$$

Then, $\mathbf{T}_\lambda$ can be calculated according to Eq. 8.

---

**Data:** Whole region $\Omega$, point trajectory $\mathcal{F}(\mathbf{p}, \lambda)$, tolerable error $\epsilon$, acceptable ratio $\alpha$, maximum iterations $N_{\text{max}}$

**Result:** The static region $\Omega_S$ and the dynamic region $\Omega_D$

initialization: $n \leftarrow 0$, $\Omega_S \leftarrow \Omega$, $\Omega_D \leftarrow \emptyset$;

**while** $n < N_{max}$ **do**

    **for** *each* $\lambda \in [0, T]$ **do**

        **Solve with L-BFGS**: $(\mathbf{R}_\lambda, \mathbf{t}_\lambda) = \underset{\mathbf{R},\mathbf{t}}{\arg\min} \|\Pi(\mathcal{F}(\Omega_S, \lambda)) - \Pi(\mathbf{R} \cdot \Omega_S + \mathbf{t})\|^2$;

    **end**

    **Compute**: $\epsilon_{max} = \max_{\mathbf{p} \in \Omega_S} \sum_{\lambda=0}^{T} \|\Pi(\mathcal{F}(\mathbf{p}, \lambda)) - \Pi(\mathbf{R}_\lambda \cdot \mathbf{p} + \mathbf{t}_\lambda)\|^2$;

    **if** $\epsilon_{max} < \epsilon$ **then**

        **stop** (solution found);

    **else**

        $\Omega_S \leftarrow \{\mathbf{p} \in \Omega \mid \sum_{\lambda=0}^{T} \|\Pi(\mathcal{F}(\mathbf{p}, \lambda)) - \Pi(\mathbf{R}_\lambda \cdot \mathbf{p} + \mathbf{t}_\lambda)\|^2 < \alpha \cdot (\epsilon_{max} + \epsilon)\}$;

        **if** $\Omega_S = \Omega$ **then**

            **stop** (solution found);

        **end**

    **end**

    $\Omega_D = \Omega \setminus \Omega_S$;

    $n \leftarrow n + 1$;

**end**

**Algorithm 1:** Static and dynamic region extraction based on trajectory analysis

---

For the motion strength, we empoly the difference between adjacent frames to replace the first-order derivative of $\mathcal{G}(\mathbf{p}, \lambda)$. Specifically, we calculate:

$$m_\lambda = \begin{cases} 0 & \text{if } \lambda = 0, \\ \frac{1}{HW} \sum_{i,j=1}^{H,W} \|\mathcal{G}(\mathbf{p}, \lambda) - \mathcal{G}(\mathbf{p}, \lambda - 1)\|_2 & \text{if } \lambda > 0. \end{cases} \tag{13}$$

Thus, we can calculate the required control signals $(\mathbf{T}_\lambda, m_\lambda)$ from any raw RGB video, which allows us to train the model with a vast array of easy-acquired RGB video data.

### 3.4 NETWORK, TRAINING AND INFERENCE

To ensure our method remains compatible with rapidly evolving base models, we have implemented an adaptive structure. Our network design is illustrated in Fig. 4. Starting from the original control signal, our adapter network generates a control feature that can be integrated into any diffusion process, thereby allowing adaptation to various video generation base frameworks.

Considering that $\mathbf{T}_\lambda$ is a 4-dim tensor with shape $(T, 2, H, W)$ and $m_\lambda$ is a 2-dim tensor with shape $(T, 1)$, we use a tiling method to expand $m_\lambda$ to the same shape as $\mathbf{T}_\lambda$, and then concatenate them along the channel dimension to finally obtain a $(T, 3, H, W)$-shaped tensor as the input of the network. As shown in Fig. 4 (layers marked with flame are our adaptive layers), we first employ several convolutional layers to convert the input to the same size as the tokens used in the diffusion process. We then concatenate the features with the tokens before computing self-attention. After the self-attention computation, we restore the original shape by removing the additional parts added during concatenation, similar to Hu (2024).

During the training phase, we merely incorporate the insertion of the control signal, while all other training strategies remain unchanged. We adopt the same loss function and the same scheduler, with the sole modification being the introduction of the control signal condition. During testing, we first employ Unidepth (Piccinelli et al., 2024) to lift the input image into a RGBD point cloud. When the user moves the camera, we convert it as the transformation of 3D points in the camera coordinate systemcan according to the camera poses. We finally project the transformed 3D points of each

frame onto camera plane to compute the 2D trajectory, as shown in Fig. 2. Additionally, the user can provide a scalar value for the motion strength control. As a result, this conforms to the training paradigm and produces suitable camera movement effects. When the motion strength is set small, nearly static camera movements can be achieved, whereas a significant motion strength allows for more pronounced dynamics of the subject.

## 4 EXPERIMENTS

In this section, we show our experiments. Sec. 4.1 introduces our implementation details and experiment settings. Sec. 4.2 shows the results and some properties of our method. In Sec. 4.3, we compare our method with previous baseline methods.

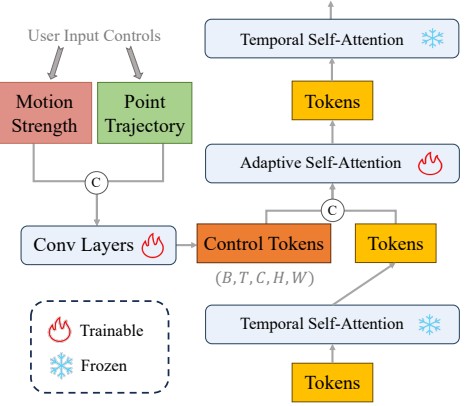

Figure 4: The adaptive network structure.

### 4.1 SETTINGS

**Implementation details.** We employ a Image-to-Video version of Magicvideo-V2 (Wang et al., 2024b) as our base model, where we set the frame number as 24 and the resolution as $704 \times 448$. We use 16 NVIDIA A100 GPUs to train them with a batch size 1 per GPU for $20K$ steps, taking about 36 hours. During training, we fix the parameters of the base model and only train our adapter part.

**Datasets.** Although previous methods trained on the RealEstate10K (Zhou et al., 2018) dataset for training, we do not choose it as our training set because the videos in this dataset are all nearly static scenes with very limited dynamic motion of objects, which is conflict with our goal of achieving controllable motion dynamics. Therefore, we collect a dataset of $30K$ video clips as our training set, which contains not only camera movements but also natural motion. For validaition, we choose two testing sets. The first testing set comprises $500$ random static scene clips from RealEstate10K, where each clip only extracts the initial frames according to the generated frame number. To enrich the testing set, we randomly substitute the camera movements in half of these clips with one of eight basic camera movements (as in MotionCtrl (Wang et al., 2023)): pan left, pan right, pan up, pan down, zoom in, zoom out, anticlockwise rotation, and clockwise rotation. The second testing set consists of $480$ samples generated by text-to-image model that feature movable objects including humans and animals, each equipped one of the basic camera movements.

**Metrics.** To comprehensively evaluate the quality of the results generated by our method and to facilitate a fair comparison with existing techniques, we employ the same metrics as in CameraCtrl (He et al., 2024): Rotational Error (`RotErr`), Translational Error (`TransErr`) and Fréchet Inception Distance (`FID`) (Seitzer, 2020). To compute `FID`, we randomly select 2000 video frames from WebVid (Bain et al., 2021). As for the calculation of `RotErr` and `TransErr`, we refer to the formula in CameraCtrl (He et al., 2024). Furthermore, as our method supports adjusting the motion dynamics, we design a metric to measure the motion dynamics in the generated videos, too. Specifically, we employ the open source RAFT (Teed & Deng, 2020) optical flow model to calculate a motion score, denoted as `MSC`. Specifically, we use optical flow to establish a correspondence relationship between any two adjacent frames, then perform 2D rigid alignment between adjacent frames (to appropriately remove the optical flow caused by camera movement), and compute the average of the $L_2$ alignment errors as the metric `MSC`.

### 4.2 VISUALIZATION RESULTS

In this section, we show the visualization results of our method, demonstrating both pixel-level controllability and the motion strength adjustment. Due to the format of the paper, the results shown below are all in frame-by-frame image format. Please see our project page for video results.

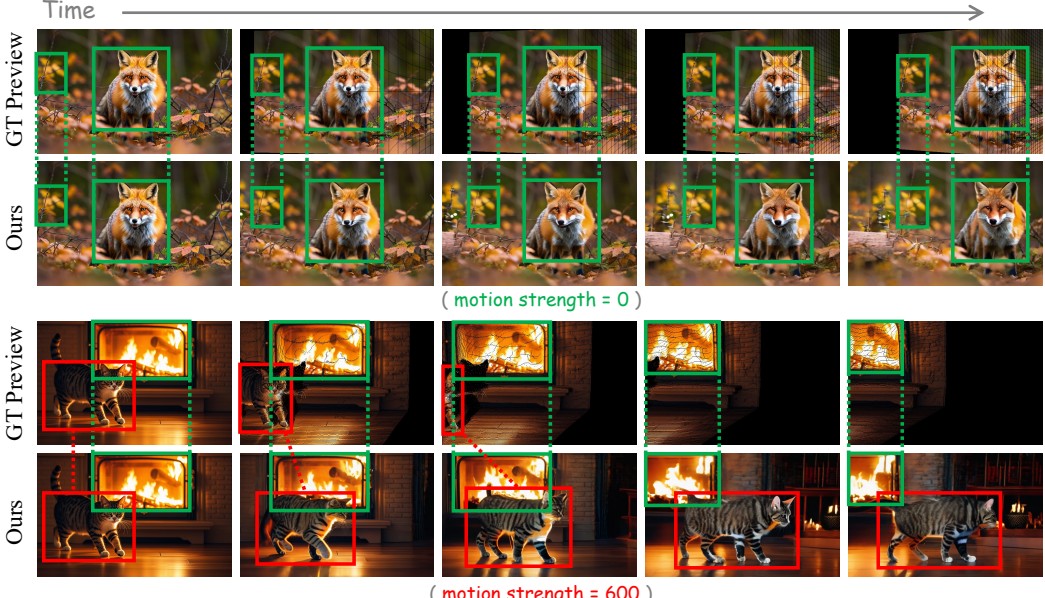

Figure 5: Visualization of our pixel-level controllability. The figure presents two samples: the top one demonstrates a pan-left camera movement, while the bottom one shows the camera sliding to the right. For each sample, we show a preview (directly render the RGBD point cloud on to 2D plane according to the extrinsic matrix) and our generated result. We can see that the generated result can almost follow the control signal at the pixel level (can be seen in the green boxes) even when there exists movable object (the cat in the red box).

### 4.2.1 PIXEL-LEVEL CONTROLLABILITY

We demonstrate comprehensive pixel-level user controllability in our approach, as illustrated in Fig. 5. Initially, we estimate the metric depth from the input image, and then directly manipulate the RGBD point cloud with control signals to render a preview image. This provides users with an immediate and intuitive visual feedback, labeled as "Preview" in the figure. Below the direct rendering results, we display the outputs generated by our framework. As observed, the camera pose in the generated results is largely consistent with the preview, indicating that our control system achieves precise pixel-level control. In the first sample, where we set the motion strength to $0$, the fox remains static, and the entire image aligns perfectly with the preview. In the second sample, with the motion strength set to $600$, the cat is able to walk on the floor. Despite the movement of cat, the camera positioning remains consistent with the preview across all static elements, such as the fireplace. These examples underscore the ability of our framework to maintain pixel-level alignment regardless of the motion strength. This high level of controllability ensures that users can interactively and effortlessly tailor their visual outputs with exceptional precision, epitomizing a truly user-friendly experience.

### 4.2.2 MOTION STRENGTH ADJUSTMENT

In Fig. 6, we illustrate the effects of varying motion strength values on the same input image with consistent camera movements. When the motion strength is $0$, the image content appears almost stationary. Conversely, as the motion strength is increased, the main objects within the scenes begin to exhibit motion. For instance, in the first example, the camera performs a pan-right movement, shifting the entire scene to the left. At a motion strength of $0$, the polar bear remains static, moving uniformly with the background. However, increasing the motion strength allows the bear to move independently, walking naturally and vividly across the frame, giving an impression of freedom and animation. In the second example, with the camera moving downward, the scene seems to ascend. With motion strength as $0$, the astronaut stands still, anchored to the ground. Increasing the motion strength causes the astronaut to walk toward the camera, thereby enhancing the dynamic interaction and realism within the scene. The third example features the camera rotating counterclockwise, which results in the scene rotating clockwise. Here, a motion strength of $0$ keeps the wolf stationary. Yet, upon intensifying the motion strength, the wolf begins to run, infusing the scene with action

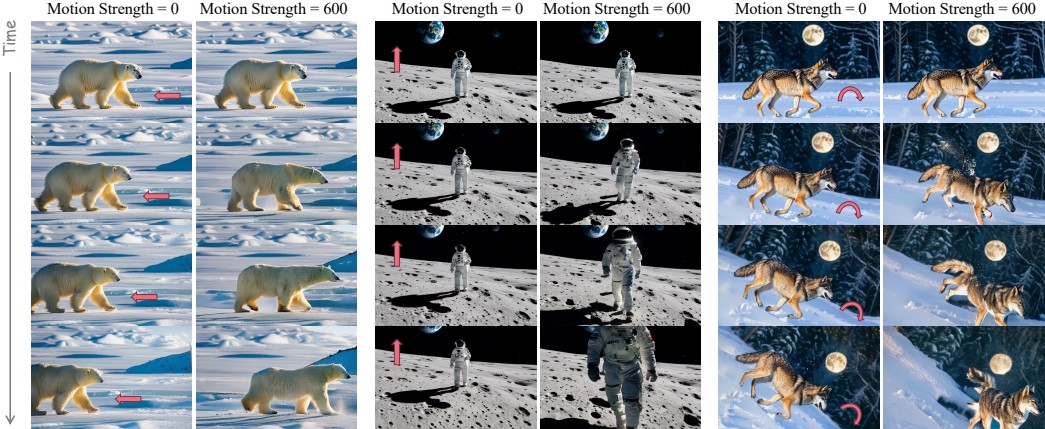

Figure 6: Results under different motion strength values. We test the same camera control signal with different motion strength value. When the motion strength is set as 0, the entire scene is nearly static even when there are movable objects in the figure (polar bear, astronaut, wolf); when the motion strength is large, the main objects moves obviously.

and enlivening the overall visualization. These demonstrations confirm the efficacy of our controlled motion strength system, showcasing its capability to customize dynamic behaviors in accordance with the desired camera movements and scene compositions.

## 4.3 COMPARISONS

In this section, we compare our results with previous baselines: MotionCtrl (Wang et al., 2023) and CameraCtrl (He et al., 2024). It is important to note that the original MotionCtrl and CameraCtrl differ significantly from our training configurations, including differences in the base model, training set, image resolution, and even the number of frames. Fortunately, they both employ an adapter architecture, allowing their designs to be adaptable to various base models. Therefore, to ensure a fair comparison, we choose to retrain MotionCtrl and CameraCtrl using the same experimental settings and base model (Magicvideo-V2) as ours. In the subsequent text of this section, whenever we refer to MotionCtrl and CameraCtrl, we are referring to the version that have been retrained by us. Considering that the motion-LoRA of AnimateDiff only supports a limited number of fixed camera movement patterns, we excluded it from our comparison.

### 4.3.1 COMPARISON ON REALESTATE10K DATASET

We compare our method with MotionCtrl and CameraCtrl on the RealEstate10K dataset. Considering that data in this dataset are nearly all static scenes, we set our motion strength to 0. Quantitative comparisons are presented in Tab. 1. Our method significantly outperforms the previous methods in both `RotErr` and `TransErr`, consistent with the pixel-level precision control observed in Section 4.2.1. For a qualitative comparison, refer to the left sample in Fig. 7. While our results are largely consistent with the preview, the outputs from CameraCtrl and MotionCtrl exhibit noticeable deviations. The results from MotionCtrl has the right trend but with some extra zoom-in, while CameraCtrl, although correct in the direction of camera movement, applies excessive movement amplitude, resulting in a failure to align at the pixel level with the ground truth. It can be seen that our method is closest to the preview image, which is consistent with the conclusion of quantitative comparison, further confirming the superiority of our controllability. Our results also show the smallest values in terms of `FID` and `MSC`, indicating that our method not only produces the highest quality of generated images but also maintains the static nature in static scenes.

### 4.3.2 COMPARISON ON DATASET OF MOVABLE OBJECTS.

We also evaluate our method against MotionCtrl and CameraCtrl on the movable object dataset. Considering it contains movable objects, we experimented with several motion strength values: 0, 200, 400, 600. Quantitative comparisons are presented in Tab. 2. Our method performs best in both

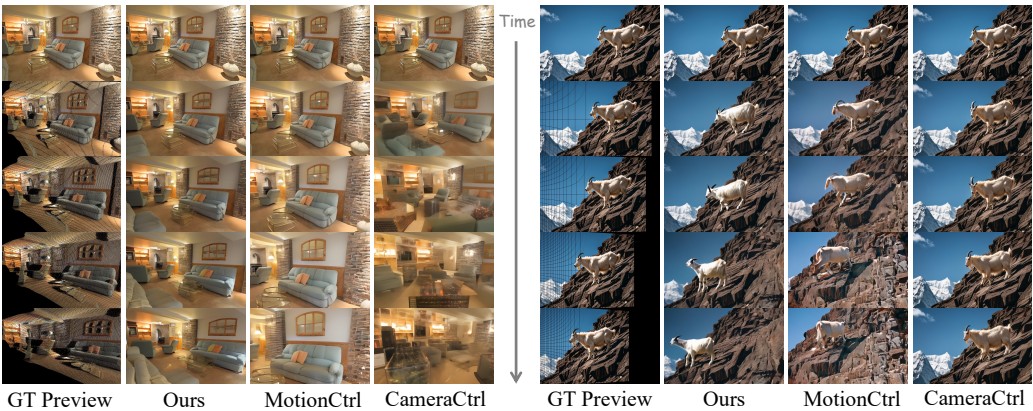

Figure 7: Qualitative comparison with previous methods. By comparing the preview with the generated results of different methods, we can see that our control precision is the best.

Table 1: Comparison on the RealEstate10k dataset.

| Methods | RotErr↓ | TransErr↓ | FID↓ | MSC↓ |
|---|---|---|---|---|
| MotionCtrl | 2.66 | 12.70 | 164.62 | 13.71 |
| CameraCtrl | 1.26 | 21.60 | 156.69 | 15.52 |
| Ours-0 | **0.53** | **9.72** | **155.01** | **12.93** |

Table 2: Comparison on the movable object dataset.

| Methods | RotErr↓ | TransErr↓ | FID↓ | MSC↓↑ |
|---|---|---|---|---|
| MotionCtrl | 2.10 | 8.08 | 98.54 | 42.28 |
| CameraCtrl | 1.56 | 11.32 | 99.59 | 32.69 |
| Ours-0 | **0.76** | **6.97** | 100.36 | **18.96** |
| Ours-200 | 1.03 | 7.53 | 93.28 | 38.23 |
| Ours-400 | 1.12 | 7.23 | 91.93 | 47.13 |
| Ours-600 | 1.18 | 8.16 | **91.86** | **47.70** |

RotErr and TransErr. Although Ours-200, Ours-400 and Ours-600 perform slightly worse on these two metrics than Ours-0, they are still better than the comparison methods. Ours-600 achieves the best FID and thus the best image quality. The FID of Ours-0 is slightly higher than that of the other settings. A possible reason for this could be that the movable objects are forcibly held static, resulting in unnatural and insufficiently diverse frames, while diversity is crucial for FID. For MSC, our smallest value (Ours-0) is lower than the comparing methods, and our largest value (ours-600) is higher than the comparing methods, which proves our adjustable motion strength control abality again. Qualitative comparison is shown on the right sample in Fig. 7, where only our method is pixel-level aligned with the ground truth.

## 5  CONCLUSION

In this work, we introduced **I2VControl-Camera**, a precise camera control method designed to enhance the controllability of video generation while maintaining a robust range of subject motion. We successfully addressed the challenge of control stability by employing point trajectories in the camera coordinate system, rather than relying on extrinsic matrices. Additionally, our approach involved modeling higher-order components of video trajectory expansion, enabling the network to precisely perceive and adjust the amplitude of subject motion dynamics. Our method demonstrated superior performance over previous methods in both quantitative and qualitative assessments. Looking forward, possible future work includes extending our framework to include more control modalities, such as drag and motion brush controls. These enhancements will allow for even more detailed and varied manipulations of video content, enabling creators to achieve a wider range of visual effects and further personalize their video productions.

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
