# OpenReview forum: "I2VControl-Camera: Precise Video Camera Control with Adjustable Motion Strength"
_ICLR.cc/2025/Conference — ICLR 2025 Poster_

### Official Review · Reviewer_whzD · 2024-10-27

**Soundness:** 3
**Presentation:** 3
**Contribution:** 2
**Rating:** 6
**Confidence:** 3

**Summary:**

This paper addresses the limitations of existing camera control methods in achieving precise control and managing subject motion dynamics in video generation. The authors introduce a novel approach, "I2VControl-Camera," which uses point trajectories in the camera coordinate system to improve control precision and stability. The method incorporates higher-order trajectory components, allowing for adjustable motion strength to achieve natural and dynamic subject movements. Experiments show that "I2VControl-Camera" surpasses state-of-the-art methods in both static and dynamic scenes, demonstrating precise control and realistic subject motion even in complex video generation scenarios.

**Strengths:**

The proposed method derives camera control precision by using point trajectories in the camera coordinate system, enhancing control tractability compared to traditional extrinsic matrix-based methods. This approach enables more accurate motion handling by explicitly modeling subject dynamics.
The method is designed to integrate with forward warp techniques and is compatible with various video generation models due to its adapter-based architecture, which maintains adaptability without modifying base model structures.
In practical applications, the method demonstrates the ability to handle deformations in complex, dynamic scenes with minimal camera movement. This adaptability allows for natural, adjustable subject dynamics and accurate control in diverse environments.
The paper is well-structured and concise, with clear figures and formulas that effectively support the proposed method's technical clarity and readability.

**Weaknesses:**

1. The authors’ approach of using point cloud poses to control video generation notably enhances control accuracy and dynamism. However, this method faces a key challenge: dense point clouds greatly increase computational cost, which impacts training stability and limits practical application. A potentially more effective control mechanism could involve using sparse control points—akin to the sparse point clouds generated by Structure-from-Motion (SfM)—which could provide a balanced trade-off between precise dynamic control and computational efficiency. I would highly encourage the authors to include a discussion comparing their method with existing approaches, particularly in terms of computational overhead, inference time, and final output quality.

2. The authors simultaneously consider 3D point motion and camera motion and utilize a unified point cloud trajectory as the control representation for video generation. However, the authors seem to only consider motion as a second-order nonlinear motion described by a Jacobian matrix in Eq.5. Compared to linear motion, what benefits does this second-order representation bring? What computational overhead does it incur? Are there further gains to be had from higher-order representations? It would make sense if the authors were to delve deeper and elucidate the reasons behind their choice of second-order representation.

3. From my perspective, the training data is expensive, and the process is redundant. It involves depth estimation and iterative L-BFGS algorithm, which are computationally costly. For complex video generation models, handling depth and solving for Rλ and tλ are expensive, limiting the algorithm's practical applicability.

4. The authors should provide a more comprehensive explanation of their implementation.

**Questions:**

1. The author should provide more details about the experimental setup, e.g., L-BFGS tolerable error, acceptable ratio, iterations, and video length of training.
2. It would be beneficial if the authors could furnish a detailed analysis of the time complexity for each component of the algorithm, as well as the memory requirements during the training process. This information is essential for assessing the algorithm's efficiency and scalability.
3. Despite the authors providing an anonymous GitHub link, it remains difficult to effectively validate the algorithm's actual effectiveness. Does the algorithm support the generation of multiple dynamic objects? What is the finest level of control achievable in terms of precision? We request that the authors provide more practical application examples to validate the algorithm's actual effectiveness, including fine-grained pixel-level control, controllable dynamic range, and sequence length of video generation. Furthermore, we ask the authors to provide additional evidence to support the claims made in the introduction section.

Currently, I would rate this paper as borderline. However, I am willing to reconsider my evaluation if the authors can provide additional clarifications and engage in a more in-depth discussion to address the concerns I have raised.

---

> ### Author Response · Authors · 2024-11-20
> **Author Rebuttal**
>
> We appreciate Reviewer whzD for the insightful feedback. Below are our responses to your comments.
>
> ---
> ## Dense control greatly increase computational cost.
>
> - Adapter Scale: Our adapter structure is notably smaller than the base model. The increased computational time with the adapter is minimal, only about 5.7% more.
>
> - Sparse vs. Dense Control Signals: Sparse control doesn't reduce the dimensionality of the tokens. MotionCtrl, for instance, uses a simple repeat method to expand signals to have (H, W) dimensions. CameraCtrl employs a plucker embedding to transform sparse signals into a dense format with shape (BatchSize, FrameNumber, ChannelNumber, Height, Width) to be compatible with attention mechanisms. Therefore, dense control does not inherently add to computational demands.
>
> - Training Complexity: Dense control signals do not increase the difficulty of training. On the contrary, by providing more motion information, they theoretically reduce the complexity of training, as they offer more direct guidance and cues to the learning process.
>
> - Sparse Feature Points from SFM: Using sparse feature points from SFM for control sounds appealing as it represents detailed information with fewer points. However, this can be challenging during user inference due to difficulties in providing such point inputs.
>
> ---
> ## Choice of second-order representation for motion strength
>
> Static parts in the world coordinate system undergo rigid/linear motions in the camera coordinate system. This linear motion is ideally represented by a linear term. Naturally, we classify the motions not accounted for by this rigid motion as being attributable to the intrinsic motion of the objects themselves. This is why we use a second-order representation to represent the object motion.
>
> ---
> ## For complex video generation models, handling depth and solving for Rλ and tλ are expensive, limiting the algorithm's practical applicability.
>
> - During the testing phase: There is no need to solve for Rλ and tλ nor to use the L-BFGS algorithm. The only requirement is invoking the Unidepth model, which incurs a negligible cost (whin 2s) relative to the overall time (about 180s) and memory consumption of the diffusion inference process.
>
> - During the training phase: Although the cost of solving offline optimization for the control signal appears high, it is relatively minor compared to the comprehensive data processing required for training the base model. Although previous methods (e.g. MotionCtrl, CameraCtrl) utilized inexpensive open-sourced data, we have demonstrated that it does not meet the requirements for our training needs. Further, once our annotated data are generated, this data can be repurposed across different base models, which is cost-effective.
>
> ---
> ## More comprehensive explanation (settings, time & memory analysis) for data pipeline implementation
> We use the pytorch implementation of L-BFGS (https://pytorch.org/docs/stable/generated/torch.optim.LBFGS.html), where we set:
>
> |    Parameter      |     Setting     |
> |-------------------|-----------------|
> |       lr          | 1               |
> |   max_iter        | 5               |
> |   max_eval        | 6.25            |
> |   tolerance_grad  | 1e-7            |
> |   tolerance_change| 1e-9            |
> |   history_size    | 100             |
> |   line_search_fn  | None            |
>
> Other parameters in Algorithn 1 :
> - $N_{max}$, $\epsilon$, $\alpha$ are set as 10, 5, 0.3
> - we only sample a grid with resolution 70 $\times$ 70 when solving L-BFGS instead of using all pixels
>
> Tips on initialization of R and t in L-BFGS : For the first time we solve {R,t} with L-BFGS, we directly derive the values from 3D point clouds as the initialization values. Subsequently, we always use the previously solved values of R and t as the initial values for the next iteration.
>
> Time complexity and memory requirement : We label data on V100 GPUs, 32G memory is sufficient. The processing time varies for different samples. On average, it takes about 40s to obtain a training sample, of which about 15s is for 3D tracking, and the rest of the time is used to solve partitions. We used 48 V100s to process the data in parallel, so the processing time was not a big obstacle. If resources are limited, I think we can reduce the resolution of the grid size in Algorithn 1.
>
> ---
> ## Additional Results and Evidence.
>
> As suggested, we have added additional visual results in our anonymous page for rebuttal(https://github.com/iclr2025sub1844/iclr2025sub1844/blob/main/rebuttal.md), including the pixel-level control, the multiple motion strength and the generation of multiple dynamic objects.

---

> > ### Comment · Reviewer_whzD · 2024-11-25
> > **revised score and code release**
> >
> > The authors' response has addressed most of my concerns, so I have revised my score to 6. I eagerly anticipate the author's decision to make their code publicly available!

---

### Official Review · Reviewer_UqgM · 2024-11-01

**Soundness:** 4
**Presentation:** 4
**Contribution:** 4
**Rating:** 8
**Confidence:** 3

**Summary:**

In this work, the authors propose a novel method to control cameras to generate a video with dynamic camera motion from a given image. The existing methods suffer from coarse controllability of cameras and difficulties to control dynamic motion of the subject, this paper resolves the aforementioned challenges by introducing point-based controlling and new training data pipeline.

From the quantitative and qualitative results provided by the authors, this paper significantly outperforms the existing methods. With the pleasant qualitative results, I think this new approach can bring the significant contribution to the community and thus, I recommend its acceptance.

**Strengths:**

1) They propose point-based control signals instead of the camera extrinsics. This method can provide dense representation to control the model to generate temporally and spatially coherent video.

2) They resolved the limitations of the in-the-wild video dataset by leveraging the off-the-shelf depth estimation (Piccinelli et al., 2024), point tracking method (Karaev et al., 2023). While this is a bit straightforward, combining with their dynamic/static segmentation, the proposed pipeline can be effective for the future training.

3) The video representation is pleasant and the paper is well written.

**Weaknesses:**

I do not observe any hard limitation to tackle.

**Questions:**

Given the proposed formulation (2), the method should be generalized to any camera motion as rotational and translational movements of the camera are function of time. But both the paper and the supplementary videos seem to include one component of the camera motion (only zoom-in/out, only rotation, only translation). I would like to hear authors' perspective of this limited results.

---

> ### Author Response · Authors · 2024-11-20
> **Author Rebuttal**
>
> We would like to thank Reviewer UqgM for the positive review. We are gratified to see the recognition of the main innovative aspects of our paper and the acknowledgment of our key contributions. Going forward, we look forward to delving deeper into the topic of video control, with the aim of further enhancing our capabilities in video motion control.
>
> ---
> ## Suggestions on combining multiple camera movements
>
> We appreciate the suggestion. As pointed out, our method indeed supports the combination of multiple camera movements, not limited to only zoom-in/out, rotation, or translation. In our anonymous page for rebuttal([https://github.com/iclr2025sub1844/iclr2025sub1844/blob/main/rebuttal.md#combinations-of-multiple-camera-movements](https://github.com/iclr2025sub1844/iclr2025sub1844/blob/main/rebuttal.md#combinations-of-multiple-camera-movements)) we have added results demonstrating various combinations of camera movements. These results vividly illustrate that our approach maintains effective pixel-level control and adjustable motion intensity even with complex camera movement combinations. We plan to feature these results on accessible platforms like our project page, where the video effects can be showcased.

---

> > ### Comment · Reviewer_UqgM · 2024-11-26
> >
> > Dear authors,
> >
> > I appreciate sharing the new results with more complex (combinations of two movements) controls of cameras. All my questions are addressed here. I will maintain my score.
> >
> > Best,

---

### Official Review · Reviewer_k3TV · 2024-11-02

**Soundness:** 3
**Presentation:** 3
**Contribution:** 2
**Rating:** 6
**Confidence:** 2

**Summary:**

This paper initially defines motion strength as the motion remaining after eliminating camera rotation and translation. Subsequently, adapter layers are incorporated and trained to adhere to the parameters of camera motion and motion strength. These parameters are extracted from captured videos through metric depth estimation, pixel tracking correspondence, and the separation of static and dynamic components via linear motion fitting. The experimental results validate the efficacy of the proposed method.

**Strengths:**

1.The definition of motion strength aligns with the concept of elastic deformation in physics, referring to the shape deformation that remains after removing rigid motion, which holds theoretical significance. The algorithm for recovering motion strength from videos is meticulously designed.

2. The quality of the generated videos is impressive. The objects in the scene exhibit natural motions, and artifacts such as flickering and inconsistent pixels are rare, even during camera movements.

**Weaknesses:**

1. Although the paper title emphasizes precise video camera control, the main body primarily discusses motion strength. The camera trajectory control capabilities of the proposed method are comparable to those of SOTA papers, like MotionLORA.

2. The camera trajectory representation is based on the projection of 3D points. Firstly, this makes the representation reliant on an additional input: 3D points. Secondly, the projection of these points onto the camera image plane results only in 2D pixel locations, meaning the projections are largely similar across different viewpoints. I am confused by the projection operation used for representing the camera trajectory.

**Questions:**

Two questions:

1. What is the range of the 2D coordinates in 3D point projection?  Are they the normalized device coordinates?
2. Which video generation model is used in the proposed method?  I did not find this information. Maybe I miss something.

**Details Of Ethics Concerns:**

No concerns.

---

> ### Author Response · Authors · 2024-11-20
> **Author Rebuttal**
>
> We would like to thank Reviewer k3TV for the valuable feedback. We are delighted to see the acknowledgment of the rationality in our definition of motion strength and are pleased with the praise for our results. Regarding some concerns, we will address them below and hope it will facilitate a better understanding of our paper.
>
> ---
> ## Which video generation model is used?
>
> As Line 340 of the paper (4 EXPERIMENTS -> 4.1 SETTINGS -> Implementation details) states, we employ a image-to-video version of Magicvideo-V2 as our base model.
>
> Furthermore, as suggested by Reviewer iTMq, we conducted additional training on a DiT-based model, Seaweed ([https://jimeng.jianying.com/ai-tool/video/generate](https://jimeng.jianying.com/ai-tool/video/generate)). The results can be viewed on our anonymous rebuttal page [https://github.com/iclr2025sub1844/iclr2025sub1844/blob/main/rebuttal.md#experiment-on-another-base-model](https://github.com/iclr2025sub1844/iclr2025sub1844/blob/main/rebuttal.md#experiment-on-another-base-model).
>
> ---
> ## The main body primarily discusses motion strength?
>
> We consider camera control and motion strength to be equally important and address both aspects in parallel; moreover, our approach also shows improvements over previous methods in terms of camera control:
>
> - From the title, we emphasized both "Precise Video Camera Control" and "Adjustable Motion Strength," which we think are both critical.
> - In section 3.1 on video representation, we refer to camera control using the rigid terms and to additional movements using higher order terms.
> - In section 3.2, we constructed two control signals: one for point trajectory, addressing camera control, and the other for motion strength, addressing the control of motion intensity. These two are designed to work in conjunction.
> - In sections 4.2.1 and 4.2.2, we discuss the good camera control capabilities and the adjustable feature of motion strength, respectively, which are both the primary focus points of our control framework.
>
> ---
> ## The camera control capability of the proposed method is comparable to those of SOTA papers, like MotionLORA.
>
> As noted in Lines 047, 144, MotionLoRA suffers from some key limitations:
>
> - Limited Flexibility: MotionLoRA methods are confined to a single fixed movement mode. For instance, a "zoom-in LoRA" cannot be used to control zoom-out movements. Even if multiple LoRAs are trained, they do not support the direct implementation of complex combinations of camera movements.
>
> - Coarse Control Scale: MotionLoRA cannot precisely control the scale of camera motion, offering only coarse control. This limitation can impact the effectiveness of the method in scenarios requiring fine-grained camera adjustments.
>
> Contrastingly, our proposed method supports arbitrary, combinable motion modes, offering significantly higher flexibility. Moreover, our approach allows for precise control over movement scales, enhancing accuracy in camera movement adjustments. This makes our method more suited for diverse and demanding applications where detailed and multifaceted camera control is essential.
>
> ---
> ## The point trajectory representation relies on an additional input: 3D points.
>
> As noted in Line 321 and Fig.2, we utilize the Unidepth method to automatically convert the input image into a 3D point cloud. This means that users are not required to provide an external 3D point cloud manually. Instead, they simply need to input the camera control and motion strength signals.
>
> ---
> ## The range of 2D coordinates? The projections are largely similar across different viewpoints？
>
> We suspect these concerns stem from our previous lack of visual demonstration for the projected trajectories. To better understand this step, we have provided visualizations of the projected point trajectories on our anonymous rebuttal homepage (see the ground truth preview). These visualizations clearly show that, following user-inputted camera control signals, there is a discernible and coherent trajectory of the projected points. Although these projected pointclouds are "largely similar," they exhibit noticeable movement, which addresses the concern about similarity across viewpoints.
>
> Regarding the range of 2D coordinates, we did not implement normalization of these coordinates (as shown in the ground truth previews), although some points may fall outside the image.

---

### Official Review · Reviewer_iTMq · 2024-11-03

**Soundness:** 3
**Presentation:** 2
**Contribution:** 3
**Rating:** 6
**Confidence:** 4

**Summary:**

The submission pintroduces a new framework for training camer-controllable text-to-video model. To achieve the goal, the authors propose to disentangle the motion into two components: one is 3D rigid transformation of the majority of the underlying 3D scene - presumably this motion component is induced by the camera movement during capturing the scene; and another component that describe the non-rigid transformation of the rest of the scene, which, on the other hand, is supposed to be induced by the movement of the subjects in the scene.

Following that, a data collection pipeline is proposed to extract associated control signals that represent the camera movement and the subjects movement of each video clip, respectively. Then a didecated adaption network is trained on top of a existing T2V model to learn a camera- and subject movement-controlled model.

**Strengths:**

The formulation that decouple the motion behind a video clip into 'global' and 'local' motions is sound.

A practical solution, that utilize several existing computer vision techniques, has been introduced to extract the associated control signals as described in that formulation.

The results produced by the proposed method demonstrate good visual quality.

**Weaknesses:**

Lack of sufficient experimental results. The submission contains a limited set of qualitative results. More video results and comparative visual results are needed to justify the effectiveness of the proposed method.

Small training dataset: The training dataset is relatively small, with only 30k video clips. While I appreciate the authors’ focus on data preparation rather than altering model structures, the limited dataset results in noticeable artifacts, such as flat character bodies in the video results. I am curious why the authors did not use a larger dataset. Is the proposed data pipeline robust and efficient enough for obtaining training data at scale?

Exposition: The paper is somewhat poorly written, leading to an unsmooth reading experience. The writing often complicates the ideas rather than simplifying them. Some expositions are even misleading. For instance, the statement “the entire 3D world can be divided into the static part and the dynamic part” at Line 161 is confusing. I believe “rigid and non-rigid motions” would be more appropriate terminology. The majority of the 3D scene remains relatively static only when the camera is fixed in the 3D scene, but the scope of this work obviously goes beyong fixed cameras.

**Questions:**

Breaking the scene motion into ‘global’ and ‘local’ movements makes sense. However, to what extent can the proposed data pipeline work? For instance, how would it handle a fly-through video over the ocean?

The point trajectory mentioned at Line 170 is confusing when it first appears. It can easily be mistaken for 3D trajectories.

The authors claim compatibility with various base models, but there are no experiments to support this claim.

How are the users supposed to provide the dense point trajectories?

---

> ### Author Response · Authors · 2024-11-20
> **Author Rebuttal**
>
> We would like to thank Reviewer iTMq for the valuable feedback. The comments and suggestions have greatly helped us in improving the quality of our work. Please see below for our responses to your comments.
>
> ---
> ## More qualitative and comparative visual results.
>
> We have provided additional qualitative and comparative visual results in our anonymous rebuttal page [https://github.com/iclr2025sub1844/iclr2025sub1844/blob/main/rebuttal.md](https://github.com/iclr2025sub1844/iclr2025sub1844/blob/main/rebuttal.md).
>
>
> ---
> ## Small training dataset.
>
> - Indeed, we did not utilize an extensive dataset for our experiments. A significant reason might be that the task of camera control does not necessarily require massive amounts of data. Initially, we began training with around 2K data samples and achieved satisfactory camera control effects. Subsequently, we expanded our dataset to 30K data points and conducted comparative experiments. We believe that the task does not require much data because it primarily involves learning the rules of pixel motion rather than generating content, which allows for good generalization.
>
> - We consider data quality to be more crucial than quantity. Although our dataset comprised only 30K training sampels, it was carefully selected to include videos with significant camera motion and substantial subject movement, ensuring the relevance and effectiveness of the data, as stated in Line 351 in the paper.
>
> - Our data pipeline is fully capable of handling large amount of data. In fact, since the submission of our paper, we have expanded our dataset to 1670K training samples and are currently training on more complex base models to achieve consumer product-level results.
>
>
> ---
> ## Writing Suggestion: "rigid and non-rigid motions" vs "static and dynamic parts"
>
> This is a good suggestion. All static objects in the world coordinate system essentially undergo a rigid motion in the camera coordinate system. The initial rationale for using the terms static/dynamic was to maintain consistency with the terminology used throughout the paper. We acknowledge the need for clarity, and we plan to address this distinction more precisely in the revised version of the paper.
>
>
> ---
> ## Writing Suggestion: "point trajectory" at Line 170
>
> We will modify the expression to indicate that we perform the point trajectory projection process.
>
>
>
> ---
> ## Effect of breaking the scene motion into ‘global’ and ‘local’?
>
> We display the visual effects of the ‘global’ and ‘local’ division results on our anonymous rebuttal page [https://github.com/iclr2025sub1844/iclr2025sub1844/blob/main/rebuttal.md#visualization-of-dynamic-mask](https://github.com/iclr2025sub1844/iclr2025sub1844/blob/main/rebuttal.md#visualization-of-dynamic-mask). We can see that our method successfully classifies subjects exhibiting substantial motion into dynamic segments with a high degree of accuracy.
>
> Following the suggestion, we have visualized some examples that feature seas or flowing water. This has proven to be an insightful experiment. From the visualized samples, it is evident that the classification of water parts into the dynamic mask is not perfectly distinct (e.g. the 3-rd sample). This imperfection might stem from the nature of water as a fluid—its movements are challenging to define, and its motion trajectories are difficult to track precisely. However, as long as there is a reasonable accuracy rate in the training data, it should sufficiently serve the training purpose.
>
>
> ---
> ## No experiments to support the claim of compatibility with various base models.
>
> - Theoretical Basis: The adaptive format inherently advantages compatibility with a variety of base models. There is no fundamental barrier to integrating our control signals into any base model for training purposes. This has also been demonstrated in previous works, such as MotionCtrl.
>
> - Experimental Validation: We conducted experiments on a new base model (Seaweed ([https://jimeng.jianying.com/ai-tool/video/generate](https://jimeng.jianying.com/ai-tool/video/generate))) to prove this compatibility. Due to limited time, the experiment is carried out at a resolution of 256x256, primarily to validate the feasibility of our approach. The results of these experiments can be viewed on our anonymous homepage [https://github.com/iclr2025sub1844/iclr2025sub1844/blob/main/rebuttal.md#experiment-on-another-base-model](https://github.com/iclr2025sub1844/iclr2025sub1844/blob/main/rebuttal.md#experiment-on-another-base-model).
>
> ---
> ## How are the users supposed to provide the dense point trajectories?
>
> As stated in Line 321 and Fig.2, we utilize the Unidepth method to automatically convert the input image into a 3D point cloud. Users can adjust 6 sliders(6 DOF, moving along xyz-directions & rotation around xyz-directions) in the interactive interface to decide the camera pose, then the camera matrix is used to compute the point trajectory.

---

> > ### Comment · Reviewer_iTMq · 2024-11-27
> > **Reviewer's response**
> >
> > Thanks for the response, which resolves most of my concerns.
> >
> > A follow-up question on the point cloud projections onto novel camera poses, I see there are unoccupied black regions in the "Input & GT Preview" columns, is it true that the network is trained with these point trajectories maps? Again, this point trajectories projection for representing cameras needs more clarification, could you please have a revised version of the paper uploaded?
> >
> > Additionally, a question may be missed. Actually, I am curious about the limitation and the associated failure examples of the method. Since the method decomposes the training video/scene into rigid and non-rigid parts, what would happen if we have videos that contain a lot of non-rigid motions, e.g., a video FULL of strong sea waves.

---

> > > ### Author Response · Authors · 2024-11-27
> > > **Author Rebuttal**
> > >
> > > 【description of point trajectory projection】
> > >
> > > We understand your concern about the description of point trajectory projection. We have uploaded a new version of PDF and strengthened the explanation of the point trajectory computation. The related description is **marked in blue**:
> > >
> > > - Figure.2 provides an intuitive display. We have reorganized the text description in the caption
> > > - In Sec.3.4, we provide the revised description on how to compute the point trajectory from the input camera movement
> > > - The Eq.(8) can clearly show the point trajectory computation
> > >
> > > We hope that the above aspects will make it easier for readers to understand our calculation process.
> > >
> > > 【is it true that the network is trained with these point trajectories maps?】
> > >
> > > As we introduced in Eq.(8) and Sec.3.4, the point trajectory we employ as control signal is the 2D coordinate of the projected points instead of the rendered images.
> > >
> > > Actually, we can render the preview image from these 2D trajectory coordinates, which shows that 2D coordinates contain more information than the rendered image. In addition, the preview image may have black areas (or black lines), which also shows its disadvantage as input.
> > >
> > > 【videos that contain a lot of non-rigid motions】
> > >
> > > The last row of https://github.com/iclr2025sub1844/iclr2025sub1844/blob/main/rebuttal.md#visualization-of-dynamic-mask shows some samples of water waves. For the 3-rd sample, we can see that some area of the water is considered as the static part and some other area is considered as dynamic, which is actually a bad case.
> > >
> > > If THE WHOLE VIDEO only contain water wave, I think that would be absolutely a bad case and the result would be somewhat random. Fortunately, there is basically no such data in our dataset because we actually filter out the data with low optical flow confidence.

---

### Author Response · Authors · 2024-11-20
**Author Rebuttal**

We sincerely thank all the reviewers for their reviews and constructive feedback. Taking into account each concern and question posed by the reviewers, we have given thorough responses within our rebuttal. It is our hope that our responses will be kindly considered during the evaluation of our submission by the reviewers and the AC. Here, we summarize the primary concerns raised by the reviewers and provide a consolidated response.

---
## More Visual Results/Comparisons (@Reviewer iTMq; @Reviewer UqgM; @Reviewer whzD)

We have provide more visual results and comparisons in our anonymous repository for rebuttal: https://github.com/iclr2025sub1844/iclr2025sub1844/blob/main/rebuttal.md.

We explain our newly added results here:

- @Reviewer iTMq : As suggested, we show [more comparative visual results](https://github.com/iclr2025sub1844/iclr2025sub1844/blob/main/rebuttal.md#pixel-level-control--visual-comparisons) in the newly added page to show the effectiveness of the proposed method.

- @Reviewer UqgM : We show [the results of multiple camera motion combinations](https://github.com/iclr2025sub1844/iclr2025sub1844/blob/main/rebuttal.md#combinations-of-multiple-camera-movements). For easier understanding, we annotate each example with the camera motion mode, such as "move left + pan right" or "move left + pan right + move up + tilt down".

- @Reviewer whzD : We show the [pixel-level controllability](https://github.com/iclr2025sub1844/iclr2025sub1844/blob/main/rebuttal.md#pixel-level-control--visual-comparisons) by previewing the ground truth control trajectory in the page. It can be seen that compared with the comparison method, our results are closest to the ground truth. We also show the results of [different motion strength](https://github.com/iclr2025sub1844/iclr2025sub1844/blob/main/rebuttal.md#multiple-motion-strength) and [multiple dynamic objects](https://github.com/iclr2025sub1844/iclr2025sub1844/blob/main/rebuttal.md#multiple-dynamic-objects).

---

### Meta-Review · Area_Chair_eFZN · 2024-12-20

**Metareview:**

The paper presents a method for allowing more precise camera control in video generation. The reviewers appreciated the approach, which decomposes scene motion into global motion, attributed to camera movement and local motion induced by objects. This decomposition enables more accurate motion handling by explicitly modeling subject dynamics. The proposed method demonstrates strong results, with natural motions and minimal artifacts. The primary weakness of the original submission was the lack of sufficient experiments. The rebuttal addressed this by providing additional experiments on pixel-level control, varying motion strength, handling multiple dynamic objects, and testing with different base models. The paper presents an effective solution for precise camera motion control in video generation, supported by impressive results.

**Additional Comments On Reviewer Discussion:**

Several issues were raised in the initial reviews, and the rebuttal effectively addressed most of them.

A primary concern is about insufficient experiments. The rebuttal effectively addressed it by providing more visual results and comparisons. Reviewers wondered how well the decomposition of scene motion into global and local ones works. The rebuttal provides experiments to show that it generally works quite well. While the proposed model claims compatibility with various base models, no experiments support the claim. The rebuttal provides experiments in which another base model is used. Additional results are provided to demonstrate that the proposed method can handle the combination of multiple camera motions. In addition, the rebuttal provides results for pixel-level control, multiple motion strength, and multiple dynamic objects per reviewers' request.

One reviewer expressed concern about the small training dataset and its potential impact on quality. The rebuttal argued that the camera control task does not necessarily require a large dataset and emphasized that data quality is more critical than quantity.

Another concern was whether the proposed method offers better camera trajectory control capabilities than existing methods like MotionLORA. The rebuttal clarified that MotionLORA provides limited flexibility and operates at a coarser control scale than the proposed method. It also highlighted that both precise camera control and motion strength adjustment are essential features offered by the proposed approach.

A reviewer requested more discussion on choosing between dense point clouds and sparse control points. The rebuttal addressed this by providing an in-depth discussion of the trade-offs and considerations involved.

The rebuttal effectively resolved most of the reviewers' concerns by providing additional experiments, detailed explanations, and further discussions, strengthening the overall contribution of the paper.

---

### Decision · Program_Chairs · 2025-01-22

Accept (Poster)